# McKittrick–Wheelock Syndrome: A Case Report

**DOI:** 10.3390/medicina59030633

**Published:** 2023-03-22

**Authors:** Kristina Marcinkevičiūtė, Marius Kryžauskas, Tomas Poškus

**Affiliations:** 1Faculty of Medicine, Vilnius University, LT-03101 Vilnius, Lithuania; 2Clinic of Gastroenterology, Nephrourology, and Surgery, Institute of Clinical Medicine, Faculty of Medicine, Vilnius University, LT-03101 Vilnius, Lithuania

**Keywords:** McKittrick–Wheelock syndrome, electrolyte imbalance, tubulovillous adenoma, villous adenoma

## Abstract

An adenoma is the most typical large bowel tumor found in 30% of all screening colonoscopies. However, it is often asymptomatic but sometimes might lead to abdominal pain or bleeding of the rectum. Critical electrolyte disbalance and acute kidney injury caused by secretory diarrhea is an untypical clinical manifestation of adenoma. It has rarely been reported in the literature and is defined as McKittrick–Wheelock syndrome. A 61-year-old patient was hospitalized for heavy dyselectrolytemia, diarrhea, acute kidney injury, sepsis, and fever. After the renal function was corrected and electrolyte imbalance persisted, visual instrumental diagnostics tests revealed a large tumor in the sigmoid colon. Subsequently, the patient underwent surgical resection, which exhibited evidence of tubulovillous adenoma on pathology. The atypical signs of McKittrick–Wheelock syndrome and comorbidities can make the diagnostics challenging. When severe hyponatremia and hypokalemia are followed by persistent mucous diarrhea, the clinicians should suspect MWS as a possible reason for it.

## 1. Introduction

An adenoma is an epithelial tumor with a glandular organization [1]. Adenomas are diagnosed in approximately one third of all performed screening colonoscopies [2]. Pathologic glandular architecture defines adenomas as tubular, villous, or tubulovillous [2]. Most villous adenomas are located in the sigmoid colon or rectum [3]. Less than 3% of all large villous adenomas tend to show hypersecretory function [4,5,6]. 

The extraordinary condition known as McKittrick–Wheelock syndrome (MWS) has been reported in the literature as electrolyte and fluid depletion related to secretory diarrhea which is caused by a large villous tumor in the colon or rectum [7]. The largest systematic study revealed a long duration of symptoms and severe electrolyte imbalances for patients with MWS [8]. However, the real prevalence of MWS in combination with colon adenoma is unknown [8,9].

In this paper, we report the case of a patient with large bowel adenoma which caused McKittrick–Wheelock syndrome.

## 2. Case Report

A 61-year-old female patient had a history of pulmonary tuberculosis in 2010. There were no signs of disease relapse, and the patient was considered recovered two years after the beginning of the treatment. The standard follow-up algorithm was used. 

The patient complained of diarrhea, bloating, and general weakness in 2019. The symptoms lasted for at least one year. Slimy watery stools were up to 6–7 times a day. General routine tests were performed, and symptomatic treatment was prescribed. The general condition of the patient improved. 

Later, the patient was treated due to acute kidney injury (AKI) and hypokalemia in 2020. A computed tomography (CT) scan was performed, which accidentally suspected a circularly thickened wall of the large bowel in the left lower quadrant. Therefore, a colonoscopy was performed. Atypical mucous membrane changes were identified, starting 20 cm from the anus, and continuing for about 30 cm long. Prominent, polypoid-shaped formations included large, confluent areas. A biopsy was taken. The lumen of the colon was slightly narrowed due to these formations and was full of continuous whitish mucus, but the endoscope could pass through it. The symptoms decreased after the conservative treatment. Unfortunately, the elective consultation with an abdominal surgeon did not take place due to the COVID-19 pandemic. 

One year later, the patient was treated for urinary tract infection and hypokalemia. Additionally, the patient complained of a dry cough, general weakness, and a lack of appetite. The recurrence of pulmonary tuberculosis was not confirmed. The conservative treatment was successful.

Nevertheless, two weeks later, the patient was presented to our hospital’s emergency department (ED) due to chest pain, shortness of breath, general weakness, nausea, diarrhea, and fever. Laboratory tests revealed high levels of urea, creatinine, deep hypokalemia, and elevated inflammatory markers, which all indicated AKI and sepsis. A chest computed tomography scan was performed due to the high level of D-dimers and acute respiratory system pathology was not identified. The patient was hospitalized in the internal medicine unit. Two hemodialyses were conducted, and kidney function was restored. Antibiotics were administered due to high inflammatory parameters. However, hypokalemia persisted despite performing hemodialysis and obtaining oral and intravenous potassium during the hospitalization period. Moreover, the patient still suffered from severe diarrhea. 

Therefore, abdominal ultrasound was performed and showed intestinal changes with the thickened wall in the pelvis. An abdominal and pelvis computed tomography scan was performed to provide more detailed information. It revealed a possible tumor mass in the sigmoid colon (Figure 1). The patient underwent a colonoscopy, which revealed a large circular adenoma in the sigmoid colon of more than 20 cm long. A biopsy was taken at the time of the colonoscopy and the histological result was tubulovillous adenoma. Internal medicine doctors suspected MWS after all the examinations and surgical treatment was suggested for the patient. 

The patient underwent a standard laparoscopic sigmoid resection with primary anastomosis. Postoperative total parenteral nutrition with bowel rest was admitted due to the high risk of anastomotic leakage for the first 7 days [10]. Antibiotic therapy was continued after the operation due to the increased level of inflammatory markers. No surgical complications occurred for the patient. Diarrhea stopped for the patient and the electrolyte imbalance was solved. No further treatment was needed. 

Macroscopically, the tumor was defined as a brownish, papillary, roughened tumor that had overgrown the mucosa (Figure 2). The final postoperative histological examination confirmed tubulovillous adenoma of the colon.

## 3. Discussion

Several examples of massive villous adenoma with severe diarrhea leading to electrolyte abnormalities have been presented in the literature since it was initially reported by McKittrick and Wheelock in 1954 [7]. Orchard et al. summarized all reported cases and found out that MWS tends to affect older female patients slightly more often, as is the case with the patient in this presented case [8]. The syndrome is typically described as the triad of acute renal failure (ARF), electrolyte imbalance, and chronic diarrhea determined by a secretory colon neoplasm [11]. MWS is more likely to be present in large adenomas where they generate significant amounts of electrolyte-containing mucin. The pathophysiology could be explained by the fact that giant adenoma has a huge surface area, which secretes more fluid than the remaining normal rectal mucosa can reabsorb, and results in chronic watery diarrhea [12]. 

MWS often includes symptoms that result from an increased fluid secretion from a rectal villous adenoma. They might be different and non-specific, and that complicates early diagnostics. The main symptoms of MWS are dehydration, mucous diarrhea, and a large variety of symptoms determined by electrolyte imbalance [13]. Patients might present with hyponatremia that may manifest as lethargy, headache, weakness, nausea, muscle cramps, and seizures, or with hypokalemia that clinically occurs in fatigue, paresthesia, cramps, ileus, vomiting, hypotension, and cardiac arrhythmias [14,15]. Moreover, it is reported that a quarter of patients with MWS have electrocardiography changes [8]. The majority of patients initially arrive with diarrhea that is often watery or mucous. Moreover, patients have a long duration of symptoms and have a history of several previous hospitalizations [12]. Depending on the size and location of the adenoma, the fluid loss might reach 3 L per day. Bicarbonate loss from the stool, typical in secretory diarrhea cases, causes metabolic acidosis [16]. In our reported case, the patient was also admitted to the hospital due to electrolyte imbalance. However, there are referred cases in the literature with an atypical presentation of MWS at the emergency department with rectal prolapse, dermatomyositis, or with encephalopathy due to severe hyponatremia [17,18]. 

The diagnostics might start with abdominal ultrasound followed by an abdomen and pelvis CT or magnetic resonance imaging (MRI). Later, a colonoscopy with biopsy for histological identification of the tumor should be performed. Because of the persistence of the symptoms and the first presumptions of other conditions, the diagnosis of MWS is frequently delayed, resulting in a severe state of volume depletion [8]. In the majority of reported cases, MWS was noticed due to electrolyte imbalance and AKI with abnormally high renal function parameters and only then was specified using visual diagnostics tests [8,19,20]. This diagnostic process was also observed in our case. The suspicion was first due to high levels of urea, creatinine, and severe hypokalemia; then, intestinal changes were noticed in abdominal ultrasound. The sigmoid colon location and the size of 20 cm of adenoma were confirmed with abdomen and pelvis CT. Consequently, tubulovillous adenoma was verified by biopsies during the colonoscopy. 

Tumor removal is the cornerstone of MWS treatment. Once renal function and electrolyte imbalance are corrected, the excision of the tumor, whether with surgery or endoscopy, should be performed [5]. Recently published studies have shown an increase in the usage of minimally invasive procedures such as transanal endoscopic microsurgery, transanal minimally invasive surgery, endoscopic mucosal resection, and endoscopic submucosal dissection [21,22]. However, some authors noticed that the risk of recurrence is greater after minimally invasive operations [4,9,18]. Endoscopic techniques may be technically challenging due to the large size of the tumor in MWS patients [18]. In our case, laparoscopic-assisted surgery was chosen. The prognosis after the removal of the tumor is excellent [23]. 

## 4. Conclusions

McKittrick–Wheelock syndrome is a rare and possibly fatal disorder. It usually presents with diarrhea, acute kidney injury, and metabolic abnormalities. Surgical treatment should be the first choice for patients with MWS.

## Figures and Tables

**Figure 1 medicina-59-00633-f001:**
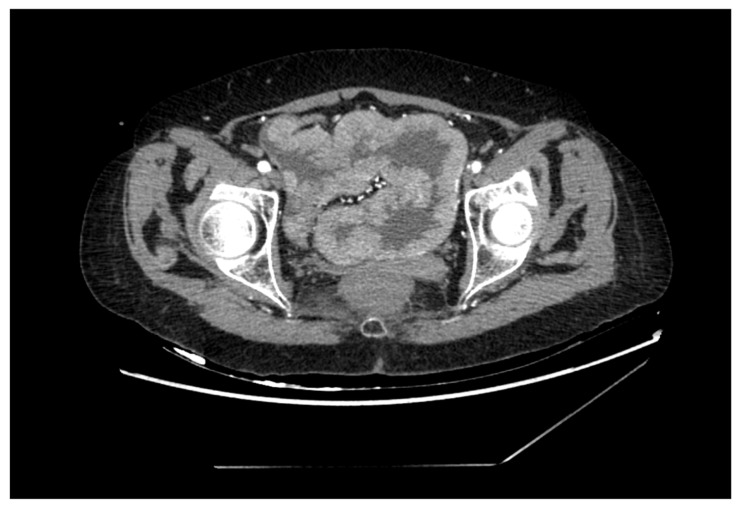
An abdominal and pelvis computed tomography: tumor mass in the sigmoid colon.

**Figure 2 medicina-59-00633-f002:**
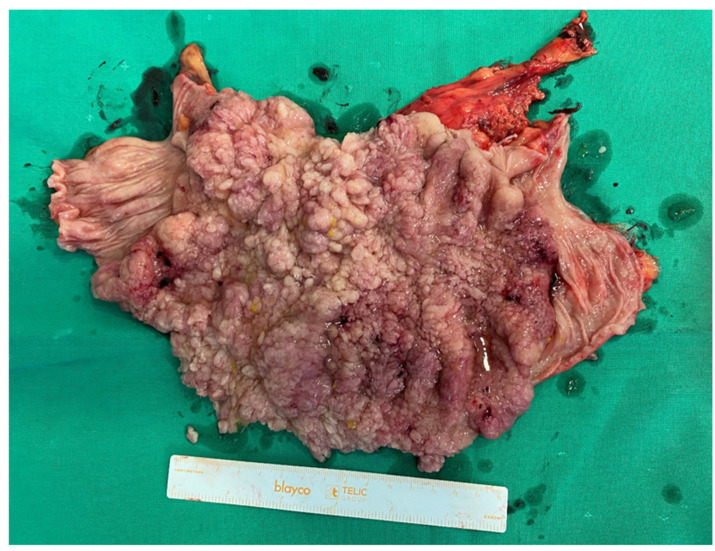
Macroscopic view of the tumor: brownish, papillary, roughened, and overgrows the mucosa.

## Data Availability

Data is contained within the article.

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
