# Peer review of "McKittrick–Wheelock Syndrome: A Case Report"

_medicina, 2023, doi:10.3390/medicina59030633_

Round 1
Reviewer 1 Report
Reports of McKittrick-Wheelock Syndrome are not uncommon in clinical work. It’s better that author can provide the results of endoscopic examination before surgery and final pathological results after operation.
Author Response
Dear reviewer,
Thank you for the comments! We provided the results of the colonoscopy and histological examination of biopsy before the operation and then the final pathological results after the operation:
The patient underwent a colonoscopy which revealed a large circular adenoma in the sigmoid colon of more than 20 cm long. A biopsy was taken at the time of the colonoscopy and the histological result was tubulovillous adenoma. <…> Macroscopically the tumour was defined as a brownish, papillary, roughened tumour that overgrown the mucosa (Figure 2). The final postoperative histological examination confirmed tubulovillous adenoma of the colon.

Reviewer 2 Report
I would like to congratulate with you for this interesting case report with literature review. The topic is interesting because of its rarity and this article could be relevant in medical practice to early detect this type of events.
The revision of the literature is appropriate.
I have no concerns about it.
I just noticed the long time interval between symptoms beginning and the diagnosis (even if you correctly outlined that it was due to the pandemic).
I suggest, at line 83, to substitute "has stopped". It would me more appropriate the past simple form "stopped". The language is on average good and the text is fluently readable.
Author Response
Dear reviewer,
Thank you for the comments! Yes, your observation is correct – it was a long interval between the first symptoms beginning and the diagnosis confirmation. That was not only due to the pandemic but also due to the nonspecific symptoms of MWS. Moreover, it might manifest with a variety of them: from symptoms determined by electrolyte imbalance (hyponatremia or hypokalemia) to electrocardiography changes or diarrhea.
Yes, your comment about the past simple tense usage is correct. We changed “has stopped” to “stopped”.
